# Growth, Development and Ornamental Value of *Miscanthus sinensis* (Andersson) Species Depending on the Dose of Shrimp Biowaste

**Piotr Żurawik**

Department of Horticulture, West Pomeranian University of Technology in Szczecin, 3 Papieża Pawła VI Str., 71-459 Szczecin, Poland; pzurawik@zut.edu.pl; Tel.: +48-91-4496366

**Abstract:** Crustaceans, including shrimps, are an important group of marine products processed in over 50 countries around the world. It is one of the most profitable and fast-growing processing branches. About 30 to 40% of crustaceans are used immediately after fishing, while 60–70% are processed. This generates thousands of tons of waste, proper management of which becomes increasingly important. The study was conducted in the years 2015–2017. Planting material included rhizomes of *Miscanthus sinensis* and *Miscanthus × giganteus*. Shrimp shells, dried and fragmented into 2–3 mm long pieces, were added to the soil at a dose of 5%, 10% and 15%. Mineral soil without the dried waste served as control. pH and substrate salinity were determined both before and after the growing season, and vegetative and generative traits of the plants were assessed. Shrimp biowaste is rich in N, P, K, Ca and Mg, has alkaline pH and high salinity. Its effects on plants depend on its dose and plant species. *Miscanthus sinensis* turned out more sensitive to the substrate salinity but in both species shrimp biowaste improved their ornamental value. For *Miscanthus sinensis* the most beneficial dose was 5%, while for *Miscanthus × giganteus* it was 15%.

**Keywords:** substrate salinity; crustaceans; waste management; grass

## 1. Introduction

Processing and consumption of seafood generate thousands of tons of waste worldwide [1–6]. Huge amounts of unprocessed biowaste impose a considerable burden on the environment [7–9]. In many countries the biowaste is thrown back into the sea where it becomes the main source of pollution in the coastal zone [1,2,10,11]. It easily dissolves in water and can be quickly absorbed by living systems [3] to support development of undesirable microflora [12,13]. This is why its proper management poses an important challenge. Present solutions involve drying [2], composting [6,14], using as animal feed [15], or agricultural fertilizers [16–18].

Considering ever stricter environmental regulations, proper management of different types of waste, including shrimp biowaste, becomes increasingly important [19]. Manufacturers also change their attitude to reprocessing and reusing the waste they generate. However, no comprehensive research has been conducted in Europe or other countries of the world investigating possible use of shrimp biowaste in the cultivation of horticultural and agricultural plants and thus creating another solution for managing this environmentally harmful waste.

The genus of *Miscanthus* comprises perennial grasses used as ornamental plants but also energy crops. However, their effective growth requires specific methods of cultivation and fertilization. Research findings on nutritional requirements of *Miscanthus* are ambiguous. According to Lewandowski et al. [20] and Kalembasa and Malinowska [21], the plants need intense fertilization,

while Jeżowski et al. [22,23] claimed that they do not need many mineral fertilizers and grow well on light soils.

In view of the above, we resolved to evaluate the possibility of using shrimp biowaste as a substrate component in the cultivation of selected *Miscanthus* species and to assess the effects of its dose on the plant growth, development and ornamental value.

## 2. Materials and Methods

### 2.1. Experimental Design

The study was carried out in the years 2015–2017 at the West Pomeranian University of Technology in Szczecin (14°31′ E and 53°26′ N). Two species of Miscanthus, *Miscanthus × giganteus* (J.M.Greef, Deuter ex Hodk., Renvoize) and *Miscanthus sinensis* (Andersson) [24] were grown in open field containers. Rhizomes were planted on 18 May 2015 and 14 May 2016 into openwork containers with a capacity of 45.0 dm$^3$. Their sides were tightly sealed with black foil, and the bottom was left open. Each container harbored five rhizomes matched for size, health and lack of mechanical damage and planted at a depth of 5 cm. The rhizomes were reproduced at the Department of Horticulture, West Pomeranian University of Technology in Szczecin.

The containers were filled with soil of loamy sand texture collected from a top layer at an experimental station WKŚIR ZUT in Szczecin located in Lipnik at Pyrzyce-Stargard Plain, 25 m a.s.l., on a border of the Płonia and the Ina basins. The substrate was enriched with dried shrimp chitin shells fragmented into 2–3 mm long pieces that are a waste product generated during shrimp processing for food purposes. The additive, called shrimp biowaste, was obtained in a single batch from a production and service company IMPROJEKT from Resko. Each year the substrate was prepared two weeks before the planned planting date that is on 4 May 2015 and 30 April 2016.The mineral soil was thoroughly mixed with the shrimp biowaste at a dose of 5%, 10% and 15% (v/v). The soil without the dried waste served as control. The substrate was used to fill the containers, and plants were grown without any fertilization.

### 2.2. Mineral Content, PH and Salt Concentration

Prior to the substrate preparation, 15 single samples of soil and shrimp biowaste were collected. They were pooled and used to determine pH in H$_2$O with a potentiometric method and CP-315M meter, and salinity with a conductometric method and CC-411 m. Our analyses also determined the content of N, P, K, Ca and Mg. Total nitrogen was determined according to Kjeldahl method following the sample mineralization in concentrated sulfuric acid and with addition of selenium mixture, total phosphorus levels were measured with colorimetric method according to Egner-Riehm, potassium and general calcium by flame photometry, and magnesium by atomic absorption spectrometry (Thermo scientific iCE 3000 series AA spectrometer) following sample mineralization in 1:1 mixture of nitric and chloric acids. pH and salinity were measured in the substrates before planting and at the end of the growing season in both years.

### 2.3. Plant Growth and Flowering

Due to variable flowering times in the first year of the study, the data on the number of days from the beginning of emergence to the beginning and completion of inflorescence development as per plant species and shrimp biowaste dose were collected only in the second year. Observations were recorded every two days.

In both years plant morphological features were assessed twice: in full bloom and at the end of the growing season. The parameters assessed included height measured from the substrate level to the highest point of the plant, number of shoots and maximum width of mature leaves. Generative features assessed in the second year of the study in fully blooming plants included the number of developed inflorescence shoots, inflorescence length from its base to the top, number of inflorescence

lateral branches and inflorescence shoot base diameter. The number of developed inflorescence shoots was also evaluated at the end of the growing season.

### 2.4. Leaf Greenness Index and Nitrogen Index

Chlorophyll Meter SPAD-502 (Minolta, Japan) was used to determine leaf greenness index in plants at their full bloom and at the end of the vegetation. The index expressed in SPAD units (Soil Plant Analysis Development) correlates with chlorophyll content [25], and may indicate plant nutritional status and possible nitrogen deficiency at consecutive growth stages [26]. Data to calculate nitrogen index were gathered on the same days with N-Tester Yara device. The measurements performed in five plants included a spot of 6 $mm^2$ located in the central part of properly developed leaves. Both indices were determined based on reading of standardized units of chlorophyll and nitrogen content in plant assimilation organs.

### 2.5. Statistical Analysis

The study investigated two *Miscanthus* species and four objects representing different doses of shrimp biowaste. All variants included three repetitions, five rhizomes each. In total, plants were grown in 24 boxes per year.

The data on the course of flowering show mean values. All statistical analyses were performed with Statistica Professional 13.3 package (TIBCO StatSoft, Palo Alto, CA, USA). The results were statistically verified by means of a one-factor analysis of variance for individual years and also collectively for two years. Mean results were compared using Tukey's test, at the significance level $p = 0.05$.

## 3. Results and Discussion

### 3.1. Mineral Content, PH and Salt Concentration

Separation of meat from the inner part of the abdomen generates post-production waste in the form of chitin shell, exoskeleton, tail and head, which, depending on the species [27], accounts for 40–50% [11,28], 45–55% [29], or up to 56% [30,31] of the animal. This study showed high content of N, P, K, Ca and Mg in the shrimp biowaste used for substrate preparation (Table 1). This is concurrent with the findings of other authors [32,33] claiming high content of macronutrients in shrimp processing waste. These minerals form the main components of the crustacean shells [29]. The content of individual minerals in shells depends on the crustacean species and fishing season [34], and also on the processing method [35]. The shrimp biowaste we used in our experiment featured high amounts of salt (Table 1). Żurawik [36] reported its lower salinity level of 28.2 g NaCl·dm$^{-3}$ in a similar study. The discrepancy was probably due to different fishing time and different crustacean species. The shrimp biowaste has alkaline pH (Table 1). This is in accordance with the publications of Evers and Carroll [37] and Rao and Stevens [31], who determined pH range between 7.4 and 8.0.

**Table 1.** Chemical properties of shrimp biowaste and mineral soil.

| Substrate Component | pH in H$_2$O | Nutrient Content (g·kg$^{-1}$) | | | | | Salt Concentration (g·kg$^{-1}$) |
|---|---|---|---|---|---|---|---|
| | | N | P | K | Ca | Mg | |
| **Shrimp biowaste** | 8.38 | 59.85 | 10.76 | 8.67 | 26.52 | 4.02 | 61.6 |
| **Mineral soil** | 6.82 | 0.84 | 0.10 | 0.07 | 2.21 | 0.14 | 0.35 |

Supplementation of the substrate with shrimp biowaste at a dose of 5% to 15% increased the substrate salinity in the beginning of *Miscanthus sinensis* and *Miscanthus × giganteus* cultivation from 1.85 to 6.79 g kg$^{-1}$ (Table 2). This was consistent with the findings of Żurawik [36], who investigated substrate supplementation with shrimp biowaste in the cultivation of *Freesia hybrida*. Dufault and Korkmaz [38] and Dufault et al. [39] also confirmed that sea bed sediments generated during crustacean

processing and involving food debris, shells and excrements enhance the substrate salinity. Substrate supplementation with shrimp biowaste elevated its pH but differences between doses of 5% to 15% were insignificant and pH reached 7.56 and 7.64, respectively (Table 2). Both pH and salinity decreased considerably during the growing season in the first and second year of the study (Table 3), which according to Kotuby-Amacher et al. [40] may affect plant growth and development.

**Table 2.** Chemical properties of the substrates before the experiment.

| Trait | Shrimp Biowaste Dose (%) | | | |
|---|---|---|---|---|
| | 0 | 5 | 10 | 15 |
| pH in $H_2O$ | 6.82 | 7.56 | 7.60 | 7.64 |
| Salt concentration (g kg$^{-1}$) | 0.35 | 1.85 | 3.11 | 6.79 |

**Table 3.** Selected chemical properties of the substrates at the end of the growing season in the first and second year of the study.

| Species | Trait | Plants | | | | | | | |
|---|---|---|---|---|---|---|---|---|---|
| | | One year | | | | Two years | | | |
| | | Shrimp Biowaste Dose (%) | | | | Shrimp Biowaste Dose (%) | | | |
| | | 0 | 5 | 10 | 15 | 0 | 5 | 10 | 15 |
| *Miscanthus sinensis* | pH in $H_2O$ | 7.00 | 7.16 | 7.19 | 7.34 | 7.15 | 7.15 | 7.19 | 7.35 |
| | Salt concentration (g kg$^{-1}$) | 0.42 | 1.75 | 2.24 | 3.31 | 0.32 | 0.74 | 0.88 | 0.84 |
| *Miscanthus × giganteus* | pH in $H_2O$ | 6.81 | 7.12 | 7.17 | 7.10 | 7.00 | 7.20 | 7.11 | 7.22 |
| | Salt concentration (g kg$^{-1}$) | 0.32 | 1.14 | 2.72 | 3.89 | 0.26 | 0.44 | 0.61 | 0.55 |

*3.2. Plant Growth and Flowering*

The available literature lacks information on the use of dried waste resulting from shrimp processing as a substrate component in the cultivation of horticultural and agricultural plants. The only published study focused on the application of shrimp biowaste in the cultivation of *Freesia hybrida* [36]. However, as the species is highly sensitive to salt, the substrate was washed twice with tap water (I—1 dm$^3$ water per 1 dm$^3$ of the substrate, II—0.5 dm$^3$ water per 1 dm$^3$ of the substrate) before beginning the experiment. Matuszak and Brzóstowicz [41] claimed that plant response to excess substrate salinity depends on the species, developmental stage, type of salt, and environmental conditions.

Our study showed different tolerance of *Miscanthus* species to substrate salinity caused by supplementation of shrimp biowaste. Our findings confirmed those reported by Chen et al. [42], who reported on considerable differences in *Miscanthus* genus species tolerance to salt. Stavridou [43] indicated that *Miscanthus × giganteus* was more tolerant to salinity than *Miscanthus sinensis*. The presented study corroborated this claim. In the first year, *Miscanthus sinensis* plants exposed to any dose of shrimp biowaste showed lower height, and developed smaller number of shoots and narrower leaves than the controls (Table 4). In *Miscanthus × giganteus* the reduction in height, number of shoots and leaf width was only visible at 10% and 15% of shrimp biowaste dose. Żurawik [36] described a similar pattern for *Freesia hybrida*. Deterioration of the plant quality at higher doses of the biowaste that greatly increased the substrate salinity, was most probably due to their reduced water uptake capacity [44], and consequently disruption of water and nutritional balance [45]. At the end of the growing season, the differences in the compared vegetative traits diminished. *Miscanthus × giganteus* plants exposed to 5% shrimp biowaste were higher, developed more shoots and wider leaves than control plants and those treated with 10% and 15% biowaste. This was concurrent with the findings of Mzabri et al. [46], who reported that contrary to high concentrations of NaCl, its small amounts positively affect growth and development of *Crocus sativus*.

Irrespective of the species, plants response to the presence of shrimp biowaste was stronger in the first than in the second year (Tables 4 and 5). Kotuby-Amacher et al. [40] also concluded that plants were more sensitive to high salinity at a seedling stage immediately after planting even though no leaf damage was found.

In the second year, the investigated species responded to the applied doses of shrimp biowaste in a different way. Increasing doses of the biowaste reduced growth of *Miscanthus sinensis* (Table 5). Substrate supplementation with low doses of bottom sediments from shrimp aquaculture and a controlled-release mineral fertilizer (Osmocote 14-6-12) enhanced rose yield in the cultivation of *Brassica oleracea* var. *botrytis italica*, while increasing doses caused excessive substrate salinity that limited the yield [35]. Evaluation a month after the first emergence showed a greater number of shoots in control variant and that with 5% biowaste and smaller in 10% and 15% variants. Consecutive measurements confirmed the most abundant shoot development in plants exposed to 5% shrimp biowaste. The same patterns occurred in the experiments with *Freesia hybrida* [36]. A month after the first emergence, control plants and those growing in the substrate with the lowest dose of biowaste developed the widest leaves. Further measurements demonstrated the greatest leaf width in plants treated with the highest dose of the biowaste. We found no significant differences in *Miscanthus* × *giganteus* height a month after the first emergences but during flowering and at the end of the growing season the plants treated with any dose of the biowaste were higher than the control ones. Despite high sensitivity of *Capsicum annuum* to salinity, Dufault and Korkmaz [38] obtained the greatest and highest quality yield at moderate doses of a sediment and a mineral fertilizer. Irrespective of the biowaste concentration, at the end of the growing season *Miscanthus* × *giganteus* plants developed more shoots and wider leaves than control plants. The differences between the studied species were most likely due to their different sensitivity to substrate salinity, as claimed by Kotuby-Amacher et al. [40].

**Table 4.** Vegetative traits of one year old *Miscanthus sinensis* and *Miscanthus × giganteus* plants depending on shrimp biowaste dose at full bloom and at the end of the growing season (mean for the years 2015–2016).

| Species | Trait | Developmental Stage | | | | | | | |
|---|---|---|---|---|---|---|---|---|---|
| | | Full Bloom Shrimp Biowaste Dose (%) | | | | End of the Growing Season Shrimp Biowaste Dose (%) | | | |
| | | 0 | 5 | 10 | 15 | 0 | 5 | 10 | 15 |
| *Miscanthus sinensis* | Height (cm) | 40.5[a] | 31.6[b] | 16.8[c] | 14.6[c] | 72.9[a] | 58.9[b] | 47.7[c] | 36.7[d] |
| | Number of shoots (pcs.) | 11.0[a] | 10.8[ab] | 9.9[ab] | 9.3[b] | 22.5[a] | 23.2[a] | 20.2[a] | 14.6[b] |
| | Leaf width (cm) | 1.4[a] | 1.4[a] | 1.3[ab] | 1.2[b] | 1.80[ab] | 1.91[a] | 1.73[b] | 1.71[b] |
| *Miscanthus × giganteus* | Height (cm) | 47.5[a] | 47.5[a] | 27.2[b] | 29.5[b] | 71.1[b] | 86.6[a] | 59.9[bc] | 48.1[c] |
| | Number of shoots (pcs.) | 11.2[a] | 11.3[a] | 12.0[a] | 11.3[a] | 26.5[b] | 37.8[a] | 16.9[c] | 16.5[c] |
| | Leaf width (cm) | 1.8[a] | 1.8[a] | 1.5[b] | 1.4[b] | 2.51[c] | 3.08[a] | 2.85[b] | 2.75[b] |

**Table 5.** Vegetative traits of two years old *Miscanthus sinensis* and *Miscanthus × giganteus* plants depending on shrimp biowaste dose one month after the first emergences, at full bloom and at the end of the growing season (mean for the years 2016–2017).

| Species | Trait | Developmental Stage | | | | | | | | | | | |
|---|---|---|---|---|---|---|---|---|---|---|---|---|---|
| | | One Month after the First Emergences Shrimp Biowaste Dose (%) | | | | Full Bloom Shrimp Biowaste Dose (%) | | | | End of the Growing Season Shrimp Biowaste Dose (%) | | | |
| | | 0 | 5 | 10 | 15 | 0 | 5 | 10 | 15 | 0 | 5 | 10 | 15 |
| *Miscanthus sinensis* | Height (cm) | 61.4[a] | 51.9[b] | 35.7[c] | 30.6[c] | 153.5[a] | 148.7[a] | 118.8[b] | 97.2[c] | 163.2[a] | 156.2[ab] | 140.7[b] | 121.1[c] |
| | Number of shoots (pcs.) | 34.9[a] | 31.6[a] | 9.6[b] | 4.7[b] | 38.8[b] | 62.0[a] | 48.1[b] | 40.7[b] | 41.3[b] | 66.8[a] | 50.3[b] | 42.0[b] |
| | Leaf width (cm) | 1.54[ab] | 1.59[a] | 1.43[b] | 1.28[c] | 1.56[c] | 1.85[b] | 1.87[b] | 2.04[a] | 1.50[c] | 1.64[c] | 1.90[b] | 2.14[a] |
| *Miscanthus × giganteus* | Height (cm) | 99.6[a] | 98.5[a] | 107.7[a] | 91.1[a] | 179.9[b] | 217.8[a] | 221.5[a] | 213.5[a] | 184.4[b] | 220.0[a] | 230.3[a] | 224.1[a] |
| | Number of shoots (pcs.) | 13.9[b] | 28.4[a] | 22.6[ab] | 12.8[b] | 18.6[c] | 38.5[b] | 51.9[a] | 45.7[ab] | 20.1[b] | 41.0[a] | 53.3[a] | 51.4[a] |
| | Leaf width (cm) | 2.08[b] | 2.42[a] | 2.42[a] | 2.18[b] | 2.21[b] | 2.65[a] | 2.73[a] | 2.78[a] | 2.20[b] | 2.64[a] | 2.77[a] | 2.66[a] |

A transition from vegetative to generative stage, i.e., to flowering, is one of the crucial stages of development [47]. Regardless of the biowaste dose, *Miscanthus sinensis* plants developed their inflorescences on average by 12.6 days earlier than *Miscanthus × giganteus* ones (Figure 1). Nuñez and Yamada [48] ascribed this discrepancy to genetic variability of the species. Delayed flowering results in prolonged vegetative stage and increased biomass yield in C4 plants [49]. Plant response to increasing doses of shrimp biowaste depended on the cultivated species. *Miscanthus sinensis* plants began flowering on average after 136.8 days of cultivation. Addition of shrimp waste delayed their flowering in a dose dependent manner. At the highest biowaste concentration, flowering began on average by 45.5 days later than in control. Żurawik [36] also demonstrated a delay in the commencement of generative stage with increasing substrate salinity with shrimp biowaste. A contrary situation occurred in *Miscanthus × giganteus*. Supplementation with shrimp biowaste at 5%, 10% and 15% accelerated flowering by 2.9, 3.4 and 7.9 days, respectively. The impact of shrimp biowaste on the course of flowering was species dependent (Figure.1). Żurawik [36] claimed the inflorescence formation time may also depend on a cultivar. *Miscanthus sinensis* responded to the presence of shrimp biowaste by prolonging the stage of inflorescence formation. In the variants treated with 5% and 10% of the biowaste this stage was longer by 17.8 and 10 days, respectively. At the highest biowaste dose the plants did not complete this developmental stage during the experiment. In *Miscanthus × giganteus* plants exposed to shrimp biowaste at 5% and 10% inflorescence formation lasted by 7.3 and 1.5 days longer, and at 15% by 12.3 days longer. According to Veatch-Blohm et al. [50], increasing substrate salinity reduced flower duration of *Narcissus* 'Dutch Master' and 'Ice Follies' by 40% to 70%.

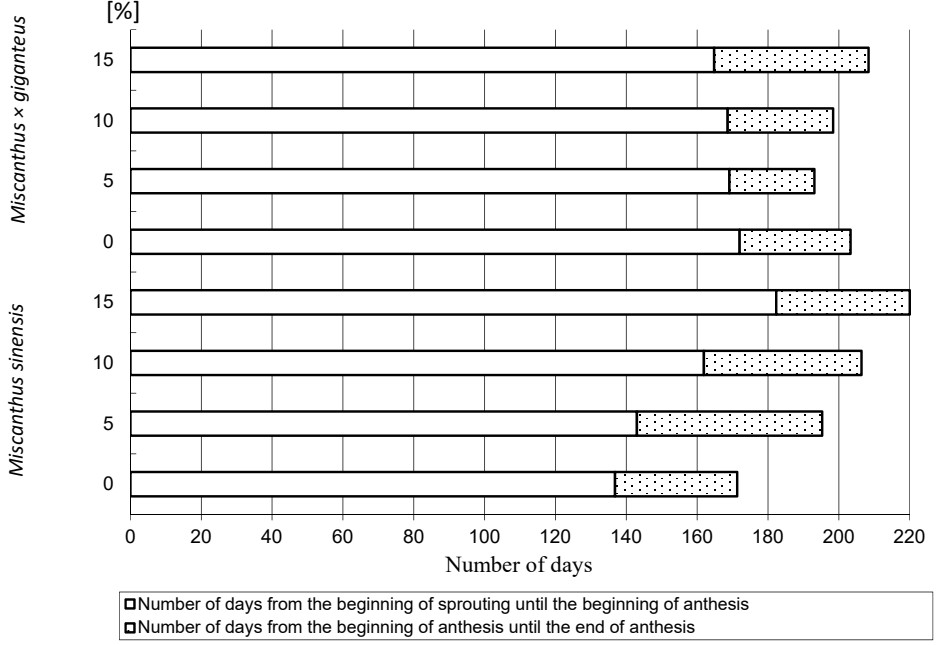

**Figure 1.** Number of days from the beginning of emergence until the end of inflorescence development in the second year of the study, depending on the species and shrimp biowaste dose (mean values for the years 2016–2017).

Ornamental value of *Miscanthus* species depended on the dose of shrimp biowaste (Table 6). In *Miscanthus sinensis* the most beneficial dose was 5%. Plants growing in such a substrate developed longer inflorescence shoots, greater number of lateral branching and the greatest number of inflorescence shoots as compared with control. Increasing the dose up to 15% deteriorated the ornamental value. Żurawik [36] described identical relationships for *Freesia hybrida*. In *Miscanthus × giganteus* the most beneficial was the highest dose of biowaste, i.e., 15%. In comparison with control, plants from this variant developed a larger number of inflorescence lateral branches and inflorescence shoots of larger

diameter. Irrespective of the dose and increasing substrate salinity, plants growing in the presence of shrimp biowaste produced more inflorescence shoots than the control ones. Veatch-Blohm et al. [50] demonstrated that substrate salinity had no effects on the number of developed flowers of *Narcissus* 'Dutch Master' and 'Ice Follies'.

**Table 6.** Flowering parameters of two years old *Miscanthus sinensis* and *Miscanthus × giganteus* plants depending on shrimp biowaste dose (mean for the years 2016–2017).

| Species | Trait | Shrimp Biowaste Dose (%) | | | |
|---|---|---|---|---|---|
| | | 0 | 5 | 10 | 15 |
| *Miscanthus sinensis* | Length of inflorescence shoot (cm) | 30.8[b] | 35.2[a] | 34.3[ab] | 33.4[ab] |
| | Number of lateral branches of inflorescence (pcs.) | 26.1[bc] | 32.2[a] | 31.1[ab] | 25.1[c] |
| | Inflorescence shoot diameter (mm) | 5.70[a] | 5.66[a] | 5.68[a] | 5.61[a] |
| | Number of inflorescence shoots (pcs.) | 35.6[b] | 51.7[a] | 24.7[c] | 9.6[d] |
| *Miscanthus × giganteus* | Length of inflorescence shoot (cm) | 34.0[a] | 34.6[a] | 34.5[a] | 36.2[a] |
| | Number of lateral branches of inflorescence (pcs.) | 42.8[b] | 48.1[ab] | 48.6[ab] | 49.4[a] |
| | Inflorescence shoot diameter (mm) | 9.06[c] | 9.82[bc] | 10.44[ab] | 10.87[a] |
| | Number of inflorescence shoots (pcs.) | 9.1[b] | 23.1[a] | 19.5[a] | 22.1[a] |

### 3.3. Leaf Greenness Index and Nitrogen Index

In the first year of the study, the addition of any dose of shrimp biowaste enhanced green coloration of *Miscanthus × giganteus* leaves both during flowering and at the end of the growing season (Table 7). A similar trend occurred in *Freesia hybrida* exposed to shrimp biowaste doses increasing from 2.5% to 15% [36]. Leaf greenness improvement confirmed the view of Pilarski [51] that plants exposed to stress increase their chlorophyll content. According to Olszewska [52], this is due to the reduction of cell size and thickening of leaf tissues that result in enhanced concentration of small-and high-molecular compounds. In *Miscanthus sinensis* increased greenness index was only visible at 5% biowaste dose. At this concentration both species showed also higher nitrogen index at all developmental stages. Majkowska-Gadomska et al. [53] reported that green coloration intensity depended on measurement date. In *Miscanthus sinensis* it was lower at the end of the growing season than during flowering. Contrary to that, Olszewska [54] observed growing greenness index in *Festuca pratensis* and *Phleum pratense* at consecutive measurement dates. We confirmed this observation in *Miscanthus × giganteus*. At the end of the growing season, all variants cultivated in the presence of shrimp biowaste showed more intense green coloration of the leaves than during flowering. Regardless of the biowaste dose, greenness index was higher in *Miscanthus × giganteus* than in *Miscanthus sinensis* leaves. Tolerant species respond to salinity by maintaining or increasing their chlorophyll content [55].

In the second year of the study, the experimental plants showed higher nitrogen index and greenness index at all measurement dates than the control ones (Table 8). Podsiadło and Jaroszewska [56] claimed that the rise in both parameters indicated an improvement in plant nutritional status. Both in *Miscanthus sinensis* and *Miscanthus × giganteus* nitrogen index was the highest in plants grown in the presence of 15% of the biowaste. The leaves of *Miscanthus sinensis* showed the most intense green coloration in plants growing in the substrate supplemented with 10% and 15% of shrimp biowaste at full bloom and at the end of the growing season. The control plants featured the lowest leaf greenness index of all variants. In *Miscanthus × giganteus* leaf greenness increased along with growing dose of shrimp biowaste. Żurawik [36] also claimed that *Freesia hybrida* plants exposed to any dose of shrimp biowaste had significantly higher greenness index at the end of the growing season than the control plants. However, this was only true for *Miscanthus* plants growing in the presence of 5% biowaste as compared with *Freesia hybrida* exposed to 15% biowaste. Kulik et al. [57] reported decreasing values of this parameter at consecutive measurements and the same tendency occurred in our study.

**Table 7.** Greenness index and nitrogen content in the leaves of one year old *Miscanthus sinensis* and *Miscanthus × giganteus* plants depending on shrimp biowaste dose at full bloom and at the end of the growing season (mean for the years 2015–2016).

| Species | Trait | Developmental Stage | | | | | | | |
|---------|-------|---------------------|---|---|---|---|---|---|---|
| | | Full bloom Shrimp Biowaste Dose (%) | | | | End of the Growing Season Shrimp Biowaste Dose (%) | | | |
| | | 0 | 5 | 10 | 15 | 0 | 5 | 10 | 15 |
| *Miscanthus sinensis* | Greenness index (SPAD) | $32.3^b$ | $35.2^a$ | $29.3^c$ | $26.4^d$ | $28.8^b$ | $35.6^a$ | $28.0^b$ | $24.4^b$ |
| | Nitrogen index | $378.6^b$ | $461.1^a$ | $356.3^b$ | $305.5^c$ | $330.0^b$ | $436.1^a$ | $335.4^b$ | $264.7^c$ |
| *Miscanthus × giganteus* | Greenness index (SPAD) | $33.4^c$ | $43.9^a$ | $38.9^b$ | $36.8^{bc}$ | $23.4^c$ | $46.8^a$ | $43.2^b$ | $43.4^b$ |
| | Nitrogen index | $412.4^c$ | $602.2^a$ | $521.6^b$ | $473.0^{bc}$ | $289.0^c$ | $599.2^a$ | $554.7^{ab}$ | $537.6^b$ |

**Table 8.** Greenness index and nitrogen content in the leaves of two years old *Miscanthus sinensis* and *Miscanthus × giganteus* plants depending on shrimp biowaste dose one month after the first emergences, at full bloom and at the end of the growing season (mean for the years 2016–2017).

| Species | Trait | Developmental Stage | | | | | | | | | | | |
|---------|-------|---------------------|---|---|---|---|---|---|---|---|---|---|---|
| | | One Month After the First Emergences Shrimp Biowaste Dose (%) | | | | Full Bloom Shrimp Biowaste Dose (%) | | | | End of the Growing Season Shrimp Biowaste Dose (%) | | | |
| | | 0 | 5 | 10 | 15 | 0 | 5 | 10 | 15 | 0 | 5 | 10 | 15 |
| *Miscanthus sinensis* | Greenness index (SPAD) | $25.0^b$ | $43.5^a$ | $44.2^a$ | $41.3^a$ | $21.0^c$ | $29.5^b$ | $34.8^a$ | $36.9^a$ | $17.7^c$ | $24.7^b$ | $32.6^a$ | $32.6^a$ |
| | Nitrogen index | $234.7^b$ | $421.2^a$ | $421.7^a$ | $391.9^a$ | $228.1^c$ | $364.6^b$ | $438.9^a$ | $463.8^a$ | $174.3^d$ | $328.5^c$ | $369.0^b$ | $403.8^a$ |
| *Miscanthus × giganteus* | Greenness index (SPAD) | $31.9^c$ | $44.2^b$ | $46.6^{ab}$ | $49.5^a$ | $23.4^c$ | $26.8^{bc}$ | $33.5^{ab}$ | $40.0^a$ | $21.0^c$ | $24.2^c$ | $33.1^b$ | $43.7^a$ |
| | Nitrogen index | $321.9^c$ | $511.7^b$ | $564.2^a$ | $555.0^a$ | $265.9^c$ | $322.1^c$ | $423.6^b$ | $542.4^a$ | $256.5^d$ | $322.2^c$ | $428.1^b$ | $535.1^a$ |

## 4. Conclusions

The waste resulting from shrimp processing is rich in such macronutrients as N, P, K, Ca and Mg, and is characterized by an alkaline pH and high salinity. Considering the above, it can be used as substrate component in the cultivation of salt tolerant *Miscanthus sinensis* and *Miscanthus × giganteus*. The biowaste doses had different effects on the investigated species. In the first year, the substrate salinity considerably limited plant height and the number of shoots, and we found *Miscanthus sinensis* to be more sensitive to such conditions. In the second year, when the salinity level dropped, biowaste presence stimulated plant growth and shoot development. It also enhanced plant ornamental value, particularly at 5% in *Miscanthus sinensis* and 15% in *Miscanthus × giganteus.* Considering a significant threat to the natural environment, further research is needed on the possibilities of using shrimp biowaste in the cultivation of horticultural and agricultural crops.

**Funding:** The study was supported by the Polish Ministry of Science and Higher Education (Project UPB 518-07-014-3171-2/18).

**Acknowledgments:** The author would like to thank Róża Stuart for her help in the laboratory.

**Conflicts of Interest:** The author declare that he has no conflict of interest.

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
