# Peer review of "Growth, Development and Ornamental Value of Miscanthus sinensis (Andersson) Species Depending on the Dose of Shrimp Biowaste"

_agriculture, doi:10.3390/agriculture10030067_

Round 1
Reviewer 1 Report
The authors mention the important challenges related to shrimp biowastes and offer an ingenious application to use it as substrate component in the cultivation of Miscanthus species.
The authors have studied, over the course of three years, the impact of different doses of shrimp biowaste mixed in mineral soils.
General comment:
While the selected measurements and experimental conditions seem appropriate to the study, data and statistical tests used to support conclusions need to be more detailed.
25. In order to benefit from better context, more details could be provided to the current global needs for substrate components for Miscanthus.
47. How many containers in total are used in the study?
49. It is understood 5 rhizomes were planted in each container two years in a row. Are the means of the different variables significantly different from each other from year over year? Please comment.
97. Please include in a new table the descriptive statistics (mean, standard deviation) for each variable, for each container available, for each year, as well as the amount of observations in each container.
100. It is not clear how Tukey test was applied, or if important factors (such as the year of growth) was taken into account. The large deviation in results
from one year to another seems to cast a shadow on the validity of the data when it comes to drawing conclusions intra-years. Please provide detailed pairwise comparisons and tests, as well as degree of freedoms and tests assumptions.
Author Response
I’m grateful for all the comments regarding my manuscript. All of them were addressed and will also be used as valuable guidelines for future publications.

Reviewer 2 Report
The paper is timely and interesting and addresses the important topic of how to manage seafood processing waste. A key point that I would like the authors to clarify is what form the shrimp shell used in the trials was in? At line 54 it mentions ‘dried shrimp chitin shells fragmented into 2-3 mm long pieces’. Please clarify if this was unprocessed shrimp shell or if some form of processing (other than drying) had been employed to concentrate or extract the chitin component. If some additional processing was involved please include a sentence or two on the likely financial and environmental costs of this. Concerning the Tables I would remove all but the essential horizontal lines. It would also aid the reader to add a quick note in the title of each Table where letters (a,b,c) are used to denote significant differences. Please use the convention 10.00 not 10,00 throughout
Minor points to address:
L48 use cm3 as opposed to dm3 throughout
L49 change to ‘open. Each … ’
L65 ‘also determined’
L68 ‘were measured with’
L74 ‘times’
L79 ‘parameters assessed’
L119 include Latin names for plants in brackets on first mention and similarly common names on first mention of Latin names throughout.
L126 ‘Table 1. Chemical …’
L128 ‘Table 2. Chemical …’
L208 ‘values’
L258 ‘Mg, and is characterised by an alkaline …’
Author Response

(The authors gave the same response as above.)

Round 2
Reviewer 1 Report
Thank you for documenting the results.